# Experimental evolution reveals a general role for the methyltransferase Hmt1 in noise buffering

**Shu-Ting You**[1,2], **Yu-Ting Jhou**[2], **Cheng-Fu Kao**[3], **Jun-Yi Leu**[1,2]*

**1** Molecular and Cell Biology, Taiwan International Graduate Program, Graduate Institute of Life Sciences, National Defense Medical Center and Academia Sinica, Taipei, Taiwan, **2** Institute of Molecular Biology, Academia Sinica, Taipei, Taiwan, **3** Institute of Cellular and Organismic Biology, Academia Sinica, Taipei, Taiwan

* jleu@imb.sinica.edu.tw

**Data Availability Statement:** Whole-genome sequence data are available at NCBI BioProject under the accession number PRJNA552720. Other relevant data can be found in the S1 Data file.

## Abstract

Cell-to-cell heterogeneity within an isogenic population has been observed in prokaryotic and eukaryotic cells. Such heterogeneity often manifests at the level of individual protein abundance and may have evolutionary benefits, especially for organisms in fluctuating environments. Although general features and the origins of cellular noise have been revealed, details of the molecular pathways underlying noise regulation remain elusive. Here, we used experimental evolution of *Saccharomyces cerevisiae* to select for mutations that increase reporter protein noise. By combining bulk segregant analysis and CRISPR/Cas9-based reconstitution, we identified the methyltransferase Hmt1 as a general regulator of noise buffering. Hmt1 methylation activity is critical for the evolved phenotype, and we also show that two of the Hmt1 methylation targets can suppress noise. Hmt1 functions as an environmental sensor to adjust noise levels in response to environmental cues. Moreover, Hmt1-mediated noise buffering is conserved in an evolutionarily distant yeast species, suggesting broad significance of noise regulation.

## Introduction

Genetically identical cells grown under homogeneous conditions can still exhibit heterogeneous phenotypes. This heterogeneity is ubiquitous and manifests at different levels, from individual protein concentrations (protein noise) [1] to cell physiology (cellular noise) [2,3]. Although phenotypic heterogeneity only exists transiently, it can lead to deterministic outcomes. In multicellular organisms, a stochastic difference in the initial cell state can result in different cell fates during development [4,5]. Moreover, stochastic variation in gene expression has been shown to determine the outcome of inherited detrimental mutations [6,7], representing a possible cause for the incomplete penetrance observed in many human diseases. In microbial cells, levels of preexisting heterogeneity can influence population fitness upon exposure to unpredictable environmental change [8,9]. This "bet-hedging strategy" is commonly

**Funding:** Funding was received from Academia Sinica (https://www.sinica.edu.tw/en; grant number AS-IA-105-L01 and AS-TP-107-ML06; to J-YL) and Taiwan Ministry of Science and Technology (https://www.most.gov.tw/?l=en; grant number 107-2321-B-001-010; to J-YL). The funders had no role in study design, data collection and analysis, decision to publish, or preparation of the manuscript.

**Competing interests:** The authors have declared that no competing interests exist.

**Abbreviations:** BFP, blue fluorescent protein; BRG1, Brahma/SWI2-related gene 1; ChIP, chromatin immunoprecipitation; DAmP, decreased abundance by mRNA perturbation; EMS, ethyl methanesulfonate; GFP, green fluorescent protein; Hsp90, heat shock protein 90; PRMT4, protein arginine N-methyltransferase 4; Q-PCR, quantitative PCR; SNP, single nucleotide polymorphism; YFP, yellow fluorescent protein.

used by microorganisms to ensure population survival without the fitness cost of developing complex regulatory networks that respond to randomly fluctuating environments [10].

At the gene expression level, preexisting cell-to-cell heterogeneity mainly originates from the stochasticity inherent to molecular processes (such as transcription factor binding to target sequences) and fluctuating levels or activities of factors critical to those processes (such as RNA polymerase II or ribosomes) [1,11,12]. Genome-wide studies have shown that low-abundance proteins often present higher protein noise, which is consistent with the greater variability of infrequent events [13,14].

However, some studies have revealed certain pathway-specific patterns in noise levels. For example, housekeeping genes tend to have lower protein noise, whereas environment-responsive genes are often noisier [15,16], perhaps because fluctuations in housekeeping genes may compromise essential cellular functions and noisy environment-responsive genes can exert a bet-hedging function. The observed patterns in these two types of genes indicate that selection operates on protein noise levels or that levels have been adjusted according to potential costs and benefits over the course of evolution [17]. Moreover, a study comparing young and old mice showed that heart cells isolated from old mice exhibit higher cellular noise than those isolated from young mice [18], suggesting that noise levels are tightly controlled in young healthy cells but that the control systems deteriorate with age.

How do cells adjust protein noise? Several general features have been associated with protein noise, including network topology, cellular compartmentalization, molecular chaperone abundance, nucleosome occupancy, and promoter architecture [7,19–22]. Genetic studies have also identified mutations that alter local or general noise levels [5,12,23,24]. Nonetheless, how cells respond to growth conditions and integrate different pathways to fine-tune protein noise remains insufficiently characterized.

To understand how protein noise is regulated, we used experimental evolution of *S. cerevisiae* to search for mutations that increased the protein noise of different reporter genes. After 35 cycles of selection, two of the evolved lines (*TDH2-GFP-* and *TYS1-GFP*-carrying lines) exhibited increased noise levels without a concomitant reduction in protein abundance. We show that increased protein noise in the evolved line carrying the *TDH2* reporter gene is not specific to the *TDH2*-related pathway, suggesting that the evolved mutations have a general effect on protein noise regulation. We identified the methyltransferase Hmt1 as the major contributor of the evolved phenotype. Further experiments revealed that noise regulation is mediated by methylation of multiple downstream targets of Hmt1 and that *HMT1* expression is often attenuated under stress conditions. Our results suggest that Hmt1 functions as a master regulator that adjusts noise levels in response to environmental stress.

## Results

### Experimental evolution of increased protein noise in budding yeast

A previous study showed that alternating selection between highest- and lowest-expression subpopulations could efficiently enrich promoter variants for high transcriptional noise in bacterial cells [25]. We hypothesized that a similar selection strategy might allow us to "evolve" yeast cells to increase the protein noise of reporter genes (Fig 1A). Eight genes (*ADK1*, *APA1*, *PCM1*, *RPL4B*, *SAM4*, *TDH2*, *TPD3*, and *TYS1*) selected from distinct cellular pathways were fused with green fluorescent protein (GFP) to generate our reporters (see Materials and methods). Evolving lines carrying individual reporter genes were subjected to alternating selection between the top 5% and bottom 5% of total populations in terms of their GFP intensity. We also treated cells with a mutagen (2.8% ethyl methanesulfonate [EMS]) before each selection cycle to increase the genetic diversity of evolving populations. After 35 cycles of selection, half

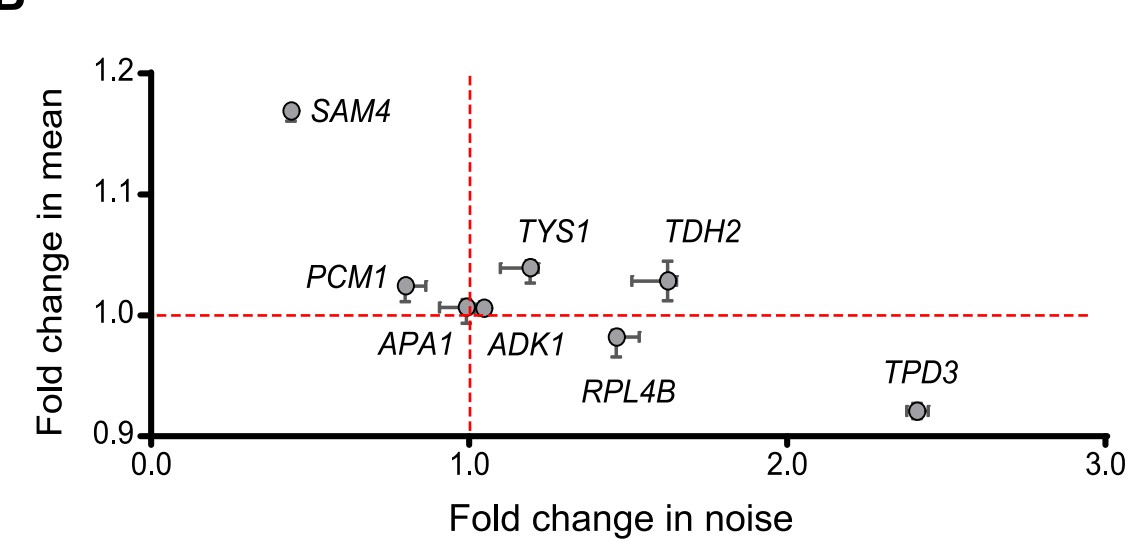

**Fig 1. Yeast cells exhibit high noise levels upon experimental evolution.** (A) Schematic of our evolution experiment. Cells were treated with 2.8% EMS, regrown for 12 h and then selected for the top 5% (or bottom 5%) of the population in terms of GFP intensity. Alternating selection enriched for mutant cells that presented higher expression noise in terms of GFP intensity. (B) Most of the evolved clones exhibit increased expression noise. Single clones were isolated from eight evolved cultures, and their reporter gene expression was measured. The x- and y-axes represent fold-change in noise and mean expression, respectively, after evolving. The median and range for 4–5 replicates for each evolved clone are indicated by circles and error bars, respectively. The red lines indicate values for the ancestral line. Data associated with this figure can be found in the supplemental data file (S1 Data). EMS, ethyl methanesulfonate; GFP, green fluorescent protein.

of the evolved lines (including the strains carrying *RPL4B*, *TDH2*, *TPD3*, or *TYS1* reporter genes) exhibited significantly increased reporter noise (Fig 1B).

Among them, the Tdh2-GFP- and Tys1-GFP-carrying lines also presented increased mean protein intensity, ruling out the possibility that the increased noise was due to reduced mean protein signal intensity. These results indicate that our selection regime effectively increased protein noise upon experimental evolution. In order to dissect the genetic basis of noise regulation, we selected the Tdh2-GFP-carrying line for further analysis because it exhibited the greatest increase in noise without a concomitant decrease in mean signal intensity.

## Isogenic cells with high and low Tdh2 levels have fitness advantages under different conditions

Bet-hedging is a commonly adopted survival strategy among microorganisms for spreading the risk of encountering hostile environments [9,26]. Tdh2 protein is a glyceraldehyde-3-phosphate dehydrogenase involved in glycolysis/gluconeogenesis, and it has been shown to help cells resist oxidative stress during the stationary phase [27]. We tested whether cells with high or low Tdh2-GFP levels represent different physiological states and whether they exhibit fitness advantages under different conditions. To do this, we isolated individual stationary-phase cells presenting different levels of Tdh2-GFP using a cell sorter and examined their phenotypes.

Consistent with a previous observation [27], upon $H_2O_2$ treatment, cells with high Tdh2-GFP levels had higher survival rates than those with low Tdh2-GFP (Fig 2A). Interestingly, when we provided fresh nutrients, the cells with high levels of Tdh2-GFP tended to reenter the cell cycle more quickly than those with low Tdh2-GFP levels, despite cells with different levels of Tdh2-GFP exhibiting no difference in survival rates under this condition (Fig 2B). This variation in cell cycle reentry is reminiscent of the divergent germination times among the individuals derived from the same population in some plants or fungi, which has been suggested to be a risk-spreading strategy to enhance long-term survival [28–30]. Divergent cell cycle reentry times can prevent an entire cell population from going extinct upon occurrence of an unpredicted environmental catastrophe. To test this hypothesis, we collected cells displaying either high or low Tdh2-GFP and challenged the cells with heat stress either before or after the cells had been re-fed with fresh nutrients. We found that cells with low Tdh2-GFP had a survival rate 3-fold greater than that of cells with high Tdh2-GFP upon encountering heat stress after nutrient refreshment. Survival rates were similar and independent of Tdh2-GFP levels for cells either nonstressed or stressed before nutrient refreshment (Fig 2C).

We also found that high and low levels of Tdh2-GFP likely represent transient states, but not genetic modifications, of the cells. When we propagated cells sorted into high- and low-Tdh2-GFP populations, Tdh2-GFP intensities of both populations reverted to a level similar to that of the initial unsorted population after a few generations (S1 Fig). Together, our results demonstrate that having cells with different levels of Tdh2 in an isogenic population is advantageous in different environments, revealing the risk-spreading benefit of expression noise.

## Increased noise is not limited to the *TDH2*-related pathway in the Tdh2-GFP evolved line

Before characterizing the detailed phenotypes of the evolved Tdh2-GFP-carrying line, we examined the cell populations and found that the signal for increased noise was not bimodal (S2A Fig). We further confirmed that Tdh2-GFP retained its full length and subcellular localization after evolution (S2B and S2C Fig) and that there were no mutations in its promoter or coding regions. These data indicate that the increased noise in the evolved line is not caused by mutations in the *TDH2* locus.

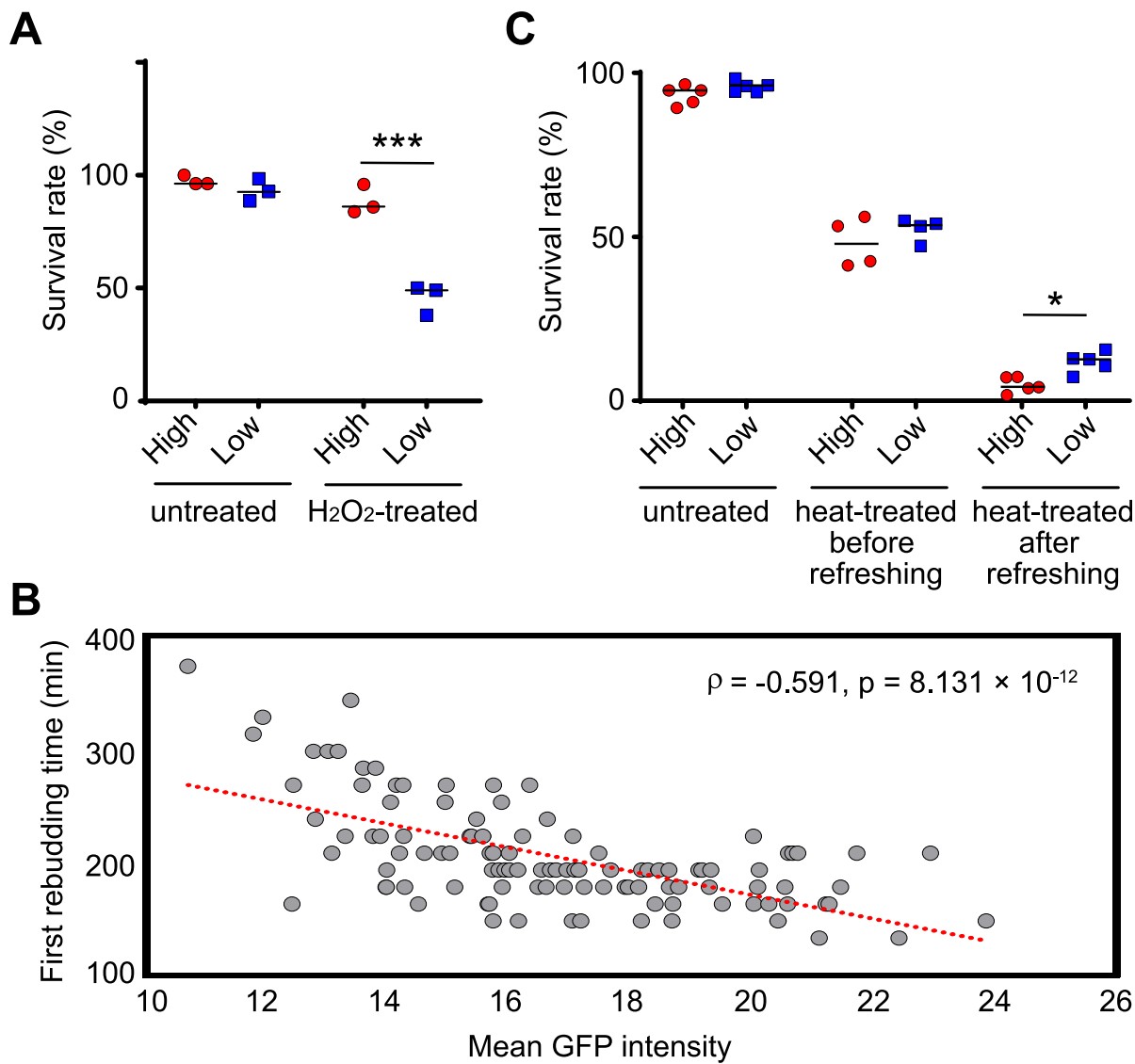

**Fig 2. Cells with different levels of Tdh2-GFP exhibit different physiological states.** (A) Stationary-phase cells with high Tdh2-GFP levels (red circles) survive better than those with low levels (blue squares) after growing in $H_2O_2$-containing medium (Fisher's exact test, $p = 0.1099$ for untreated samples, $p < 2.2 \times 10^{-16}$ for $H_2O_2$-treated samples). Single cells with high or low GFP intensities were sorted and plated on the same plates with or without 4.4 mM $H_2O_2$. Survival rates were determined by counting colony-forming units after 5 d of growth. (B) Stationary-phase cells with high Tdh2-GFP tend to reenter the cell cycle faster than those with low Tdh2-GFP signal. Each dot of the scatterplot represents data from a single cell. Unsorted stationary-phase cells were placed on YPD agarose pads, and only unbudded cells were monitored using time-lapse microscopy. The x-axis indicates the initial Tdh2-GFP signal intensity for each cell, and the y-axis indicates the first rebudding time. The red dotted line represents a linear regression (Spearman's rank correlation, $n = 111$, $p = 8.131 \times 10^{-12}$). (C) Stationary-phase cells with low Tdh2-GFP signal survive better than those with high signal when cells encounter heat stress after being re-fed with fresh nutrients (one-sided Wilcoxon rank-sum test, $n = 4$–5; $p = 0.1995$ for untreated samples, $p = 0.2426$ for heat-treated samples before nutrient refreshment, $p = 0.0159$ for heat-treated samples after nutrient refreshment). Survival rates were determined by counting colony-forming units after 3 d of growth. The median value for replicates is indicated by solid horizontal lines among groups of data points. $^*p < 0.05$, $^{***}p < 0.001$. Data associated with this figure can be found in the supplemental data file (S1 Data). GFP, green fluorescent protein; YPD, 1% yeast extract, 2% peptone, and 2% dextrose.

If the evolved mutations in the Tdh2-GFP-carrying line occurred within sequences pertaining to general noise regulators, these mutations may affect both the *TDH2*-related pathway and other pathways. We selected four pertinent genes—*TDH3*, *PGK1*, *ADK1*, and *GLY1*—to investigate the effect of the evolved mutations. *TDH3* is a paralog of *TDH2* that encodes an

enzyme for glycolysis/gluconeogenesis. *PGK1* encodes another key enzyme involved in glycolysis/gluconeogenesis. Both *TDH3* and *PGK1* are likely coregulated with *TDH2* because all three respective proteins operate in the same metabolic pathway [31]. *ADK1* encodes an adenylate kinase required for purine metabolism [32], and *GLY1* encodes a threonine aldolase involved in glycine biosynthesis [33], neither of which is related to the *TDH2* pathway. Nonetheless, all four genes exhibited increased protein noise in the evolved line (Fig 3A), suggesting that the effect of the evolved mutations is not pathway-specific.

We also subjected a *TDH2* promoter-driven GFP construct to the same analysis and observed increased noise in the respective evolved line (Fig 3B), raising the possibility that increased noise in the evolved cell lines may be attributable, at least in part, to promoter regulation or mRNA turnover.

## An Hmt1 methyltransferase mutant significantly contributes to the evolved increase in noise

To understand the genetic basis of the increased noise we observed in the evolved Tdh2-GFP-carrying line, both ancestral and evolved lines were subjected to whole-genome sequencing. A total of 1,022 mutations (including 494 nonsynonymous, 256 synonymous, and 271 intergenic mutations) were identified in the evolved genome (S1 Table). This high number of mutations is most likely due to the mutagen treatment we applied during the cycles of selection.

Next, we used bulk segregant analysis to refine the list of candidate mutations. To do this, we crossed the evolved line to the ancestral line and analyzed their F1 haploid progeny. We measured the noise level of Tdh2-GFP in 360 segregants and established an "evolved-like" pool (comprising 16 segregants) and an "ancestral-like" pool (20 segregants; see S3 Fig and Materials and methods). Both of these segregant pools were then subjected to whole-genome sequencing. Based on our computational simulation, we applied two criteria to select candidate mutations from these segregant pools: (1) the mutation frequency in the "evolved-like" pool had to be >70%; and (2) the difference in mutation frequencies between the "evolved-like" and "ancestral-like" pools was >38% (see Materials and methods). Twenty nonsynonymous mutations met these two assumptions and were subjected to reconstitution experiments (S2 Table).

We first introduced the candidate mutations into the ancestral line using the CRISPR/Cas9 system and then examined the noise level of Tdh2-GFP. Among the tested reconstitution lines, we found that a mutation (G70D) in the methyltransferase-encoding gene, *HMT1*, resulted in a significant increase of noise, close to the level exhibited by the evolved line (Fig 4A). Similarly, we observed increased noise in the *TDH2* promoter-driven GFP construct (Fig 4B). When we rescued the mutation in the evolved line by reverting to the wild-type sequence, Tdh2-GFP noise was reduced to a level similar to the ancestral line (Fig 4A). These data indicate that *hmt1-G70D* is a primary mutation contributing to the evolved increase in noise.

Already in this study, we have shown that proteins in both *TDH2*-related and *TDH2*-unrelated pathways exhibited increased noise in the evolved line (Fig 3A). In the *hmt1-G70D* reconstitution line, we observed similar noise increases in the *TDH3*, *ADK1*, and *GLY1* reporter genes (Fig 4C), indicating that the increased noise observed in the evolved line is mainly due to the *hmt1-G70D* mutation.

## The noise buffering effect of Hmt1 is mediated through multiple methylation targets

Hmt1 is a methyltransferase that methylates arginine residues in its substrates. The mutated G70D residue is located within a conserved methyltransferase motif (S4A Fig), and mutations in this motif cause loss of methyltransferase activity in *Escherichia coli* and yeast [34,35]. The

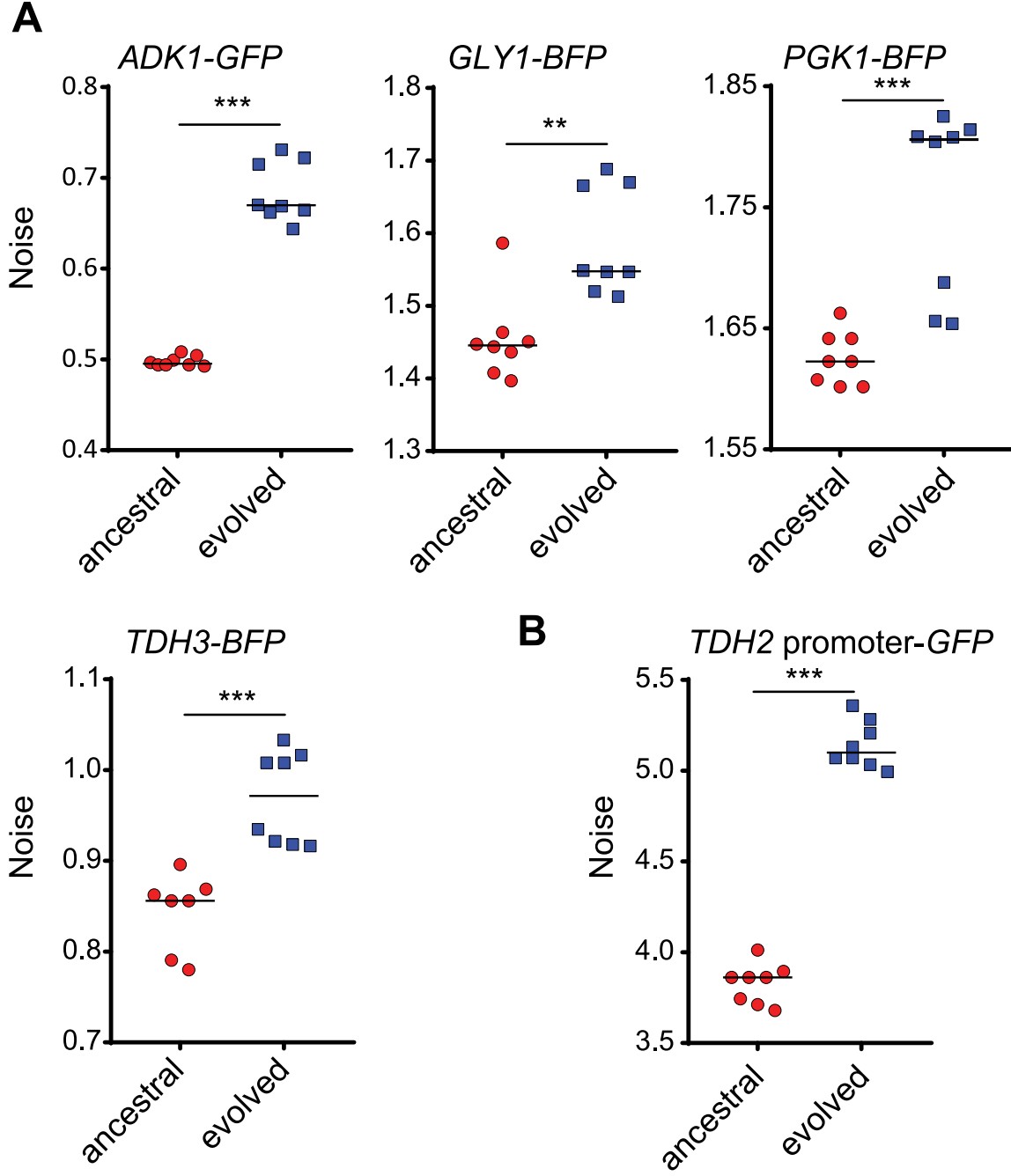

**Fig 3. Multiple genes in the evolved cells exhibit increased noise levels.** (A) Four reporter genes (*TDH3-BFP*, *PGK1-BFP*, *ADK1-GFP*, and *GLY1-BFP*) were engineered in modified ancestral (red circles) and evolved (blue squares) *TDH2-GFP*-carrying cells (see Materials and methods), and their expression noise was measured (one-sided Wilcoxon rank-sum test, $n = 7$–8; $p = 4.534 \times 10^{-4}$ for *ADK1*, $p = 0.0027$ for *GLY1*, $p = 9.441 \times 10^{-4}$ for *PGK1*, $p = 7.158 \times 10^{-4}$ for *TDH3*). *TDH3* and *PGK1* are involved in *TDH2*-related pathways, whereas *ADK1* and *GLY1* are not. (B) Transcriptional regulation is responsible for increased noise in the evolved cells (one-sided Wilcoxon rank-sum test, $n = 7$–8; $p = 4.495 \times 10^{-4}$). The *TDH2* promoter was directly fused with the coding sequence of *GFP*. This construct was engineered in modified ancestral and evolved *TDH2-GFP*-carrying cells, and its expression noise was measured. The median value of replicates is indicated by horizontal solid lines among groups of data points. $^{**}p < 0.01$, $^{***}p < 0.001$. Data associated with this figure can be found in the supplemental data file (S1 Data). BFP, blue fluorescent protein; GFP, green fluorescent protein.

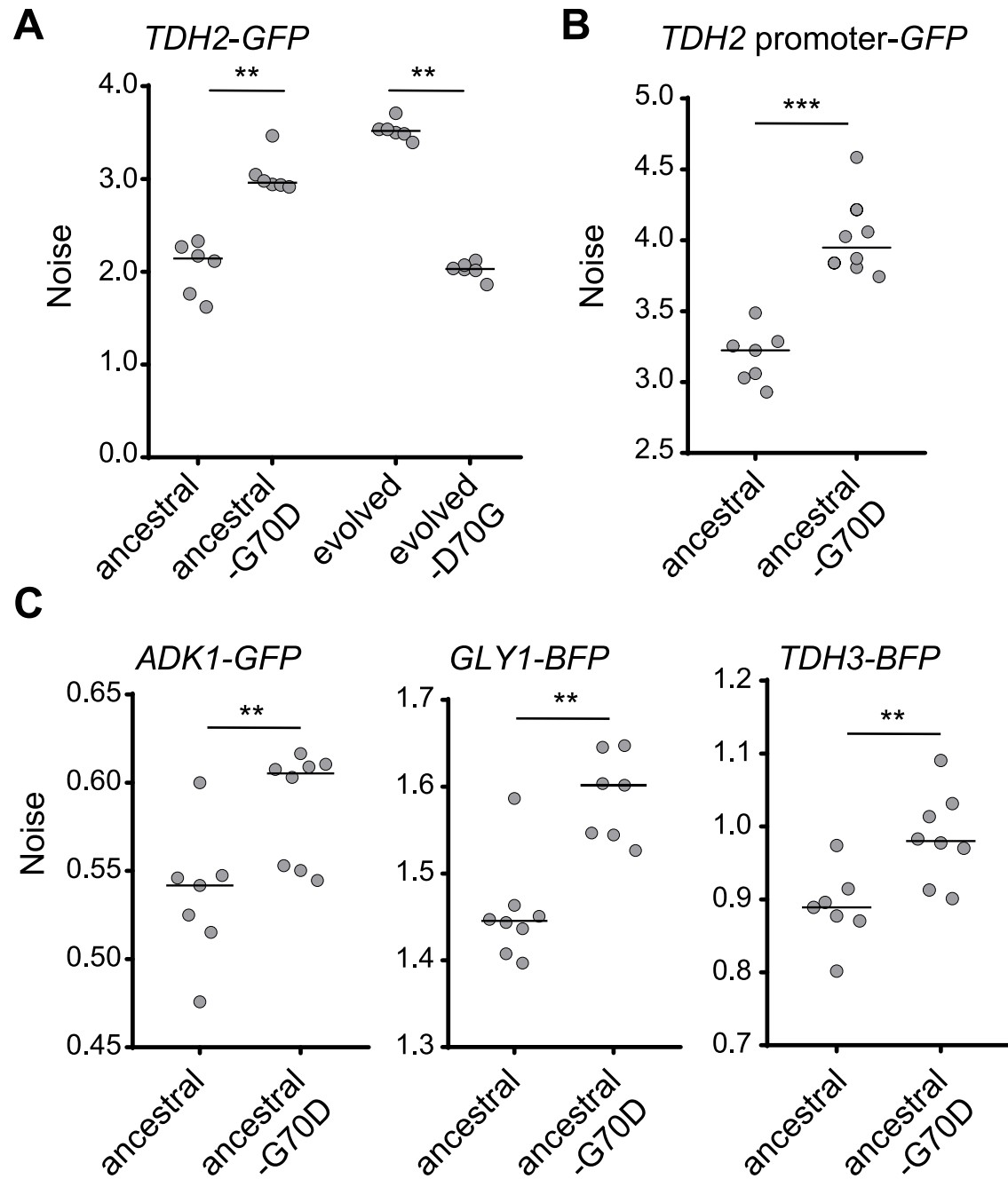

**Fig 4. The *hmt1-G70D* mutant recapitulates the increased noise phenotype observed in the evolved line.** (A) Reconstituting the *hmt1-G70D* mutation in the ancestral background increases Tdh2-GFP noise, whereas reversing that mutation in the evolved background decreases it (one-sided Wilcoxon rank-sum test, *n* = 6; *p* = 0.0011 for the ancestral background, *p* = 0.0025 for the evolved background). (B) Reconstituted *hmt1-G70D* mutant cells exhibit increased expression noise of the *TDH2* promoter–GFP construct (one-sided Wilcoxon rank-sum test, *n* = 8; *p* = 0.0002). (C) Reconstituted *hmt1-G70D* mutant cells present increased noisy expression of Adk1-GFP, Gly1-BFP, and Tdh3-BFP (one-sided Wilcoxon rank-sum test, *n* = 7–8; *p* = 0.0030 for *ADK1*, *p* = 0.0011 for *GLY1*, *p* = 0.0030 for *TDH3*). All mutants were constructed using the CRISPR/Cas9 system. The median value of replicates is represented by horizontal solid lines among groups of data points. **p < 0.01, ***p < 0.001. Data associated with this figure can be found in the supplemental data file (S1 Data). BFP, blue fluorescent protein; GFP, green fluorescent protein.

*hmt1-G70D* mutant also exhibited a noise increase similar to levels exhibited by *HMT1*-deletion cells (S4B Fig). Moreover, western blot analysis using anti–methylated arginine antibodies confirmed that *hmt1-G70D* mutant cells lack Hmt1 methyltransferase activity (S4C Fig). These results suggest that the noise-regulating function of Hmt1 is mediated through its methyltransferase activity.

In vivo and in vitro studies have identified several Hmt1 substrates [36–38]. Moreover, Hmt1-mediated methylation has been shown to enhance the function of its substrates in multiple pathways, including chromatin remodeling and transcription [39–41], translation and ribosome biogenesis [42–45], and posttranscriptional regulation [46–49]. We selected the representative proteins Npl3, Rps2, Sbp1, and Snf2 from among these pathways and generated corresponding deletion or hypomorphic mutants (in cases when mutant haploids died or had severe growth defects) and investigated noise levels among these mutants.

Of all these tested mutants, only the *rps2* and *snf2* mutants presented significantly increased noise (Fig 5A), with the other mutants showing either no change or reduced noise (Fig 5B). Rps2 is a component of the small ribosomal subunit, suggesting involvement of translational regulation in noise control. Snf2 is the catalytic subunit of the SWI/SNF chromatin remodeler, and its function depends on two other SWI/SNF components, Snf5 and Snf6 [12,50,51]. We further assessed *snf5* and *snf6* mutants and found that they also presented increased noise (Fig 5A and 5C), confirming the general role played by the SWI/SNF chromatin remodeler in noise regulation.

To test whether *hmt1-G70D* mutation affects the activity of SWI/SNF complexes, we performed a functional assay of the SWI/SNF chromatin remodeler. It has been shown previously that transcriptional activation of the *CHA1* gene is attenuated when cells have defective SWI/SNF complexes [52]. We monitored mRNA levels of *CHA1* immediately after shifting cells to a *CHA1*-inducing medium. Expression of *CHA1* was indeed reduced in *hmt1-G70D* mutant cells and in *snf5* mutants (Fig 5D), indicating that the function of SWI/SNF complexes is compromised in *hmt1-G70D* mutant cells.

We then examined the promoter regions of *TDH2* and *GLY1* (unrelated to *TDH2*) by histone chromatin immunoprecipitation (ChIP) combined with quantitative PCR (Q-PCR) to understand how Hmt1 influences gene expression. We observed increased nucleosome occupancy in *hmt1-G70D* mutant cells (Fig 5E), providing evidence for the supposition that noise regulation partially operates via SWI/SNF-mediated nucleosome remodeling [21].

## Hmt1 functions as a mediator in responses to environmental stresses

Genome-wide studies have indicated that *HMT1* expression frequently fluctuates under different growth conditions [53,54], suggesting that cells may employ an environmental sensor to enhance phenotypic heterogeneity upon encountering stress. We measured the mRNA levels of *HMT1* under various stress conditions—including heat, oxidative stress, high osmolarity, and glucose starvation—and found that they were significantly reduced under all of these conditions (Fig 6A). Consistent with the concept of an environmental sensor, the noise level of Tdh2-GFP was also increased under stress conditions (Fig 6B and S5 Fig). Together, these data indicate that Hmt1 can serve as a mediator to control levels of phenotypic heterogeneity in response to environmental stimuli.

Our data show that bet-hedging within a population could be represented by the noisy expression of Tdh2-GFP (Fig 2). If increased noise can help populations survive stressful environments, we anticipated that cells with low Hmt1 activity under stress would present an enhanced survival rate relative to those with normal Hmt1 activity under stress. Indeed, when

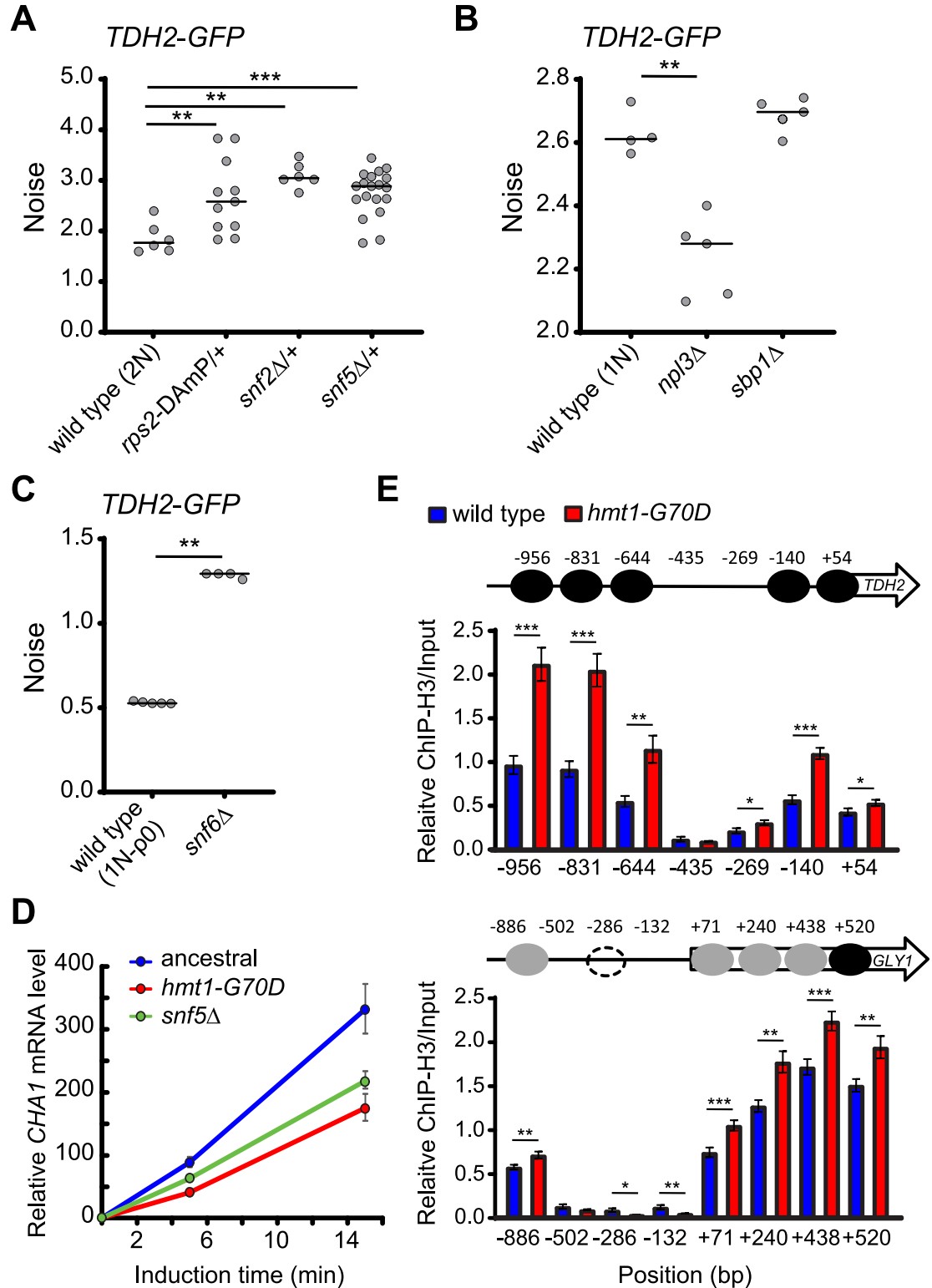

**Fig 5. Mutations of SWI/SNF components and the small ribosomal subunit Rps2 result in elevated noise.** (A) Attenuating *RPS2*, *SNF2*, or *SNF5* gene expression in the ancestral background results in increased noise (one-sided Wilcoxon rank-sum test, $n = 6$–$19$; $p = 0.0039$ for *rps2*-DAmP/+, $p = 0.0025$ for *snf2*Δ/+, $p = 0.0008$ for *snf5*Δ/+). Rps2 and Snf2 are methylation substrates of Hmt1. Snf2, Snf5, and Snf6 are essential components of the SWI/SNF chromatin remodeler. Since haploid mutants of *rps2*-DAmP, *snf2*Δ, and *snf5*Δ exhibit severe growth defects and because slow growth has been shown to increase expression noise [91],

we constructed heterozygous mutant diploids to measure noise. (B) Deletions of *NPL3* or *SBP1* in haploid cells do not result in increased noise (one-sided Wilcoxon rank-sum test, *n* = 5; *p* = 0.0079 for *npl3Δ*, *p* = 0.206 for *sbp1Δ*). Npl3 and Sbp1 are methylation substrates of Hmt1. (C) Deleting *SNF6* in the ancestral background results in increased noise (one-sided Wilcoxon rank-sum test, *n* = 5; *p* = 0.0087). Haploid *snf6Δ* mutants exhibit defective respiration, so we used a haploid rho- ancestral strain in this experiment, and cells were grown in YPD medium. The median value of replicates is represented by horizontal solid lines among groups of data points. (D) The activity of SWI/SNF chromatin-remodeling complexes is compromised in the *hmt1-G70D* mutant. It has been shown previously that compromising SWI/SNF complexes results in attenuated transcriptional activation of *CHA1* [52]. Total RNA was isolated from ancestral, *hmt1-G70D*, and *snf5Δ* haploid cells 0, 5, and 15 min after adding 0.1% L-Serine (an inducer of *CHA1* expression). Specific mRNA levels were assessed by Q-PCR. In the figure, *CHA1* mRNA levels were normalized to those of *PYK1*, with this latter acting as an internal control. For all time points, *CHA1* mRNA levels in ancestral cells (blue) were significantly higher than those in *hmt1-G70D* (red) or *snf5Δ* (green) mutant cells (one-sided Student's *t* test, *n* = 9; *p* < 0.01). (E) *hmt1-G70D* mutant cells display higher nucleosome occupancy in the promoter and partial coding regions of *TDH2* and *GLY1*. The chromatin status of the *TDH2* promoter was established by ChIP against H3 coupled with Q-PCR. The numbers in the x-axis indicate the distance (in bp) from the transcription start site. The y-axis represents relative enrichment of H3 signals for the amplicons at the indicated regions (one-sided Student's *t* test, *n* = 10–12). Black, gray, and dashed circles indicate confirmed, fuzzy, and condition-specific nucleosome-occupied regions, respectively. Error bars represent standards errors. *$p$ < 0.05, **$p$ < 0.01, ***$p$ < 0.001. Data associated with this figure can be found in the supplemental data file (S1 Data). ChIP, chromatin immunoprecipitation; DAmP, decreased abundance by mRNA perturbation; *GFP*, green fluorescent protein; H3, histone 3; Q-PCR, quantitative PCR; YPD, 1% yeast extract, 2% peptone, and 2% dextrose.

we treated wild-type and *hmt1-G70D* mutant cells with $H_2O_2$, the mutant population exhibited better viability (Fig 6C).

## Hmt1-mediated noise buffering is conserved in *S. pombe*

Our results suggest that the methyltransferase Hmt1 may function as a general noise regulator that constrains physiological noise in normal conditions but facilitates heterogeneity under stress. This buffering mechanism is likely to increase long-term population survival. Methyltransferase is a conserved enzyme that exists in all eukaryotic kingdoms. We examined whether its buffering function is also conserved in a phylogenetically distinct species, *S. pombe*, which diverged from the common ancestor of *S. cerevisiae* at least 400 million years ago [55]. We generated yellow fluorescent protein (YFP) fusion protein constructs of *tdh1* and *gpd3* (the *S. pombe* orthologs of *TDH2*) and examined their noise levels in the wild-type and *rmt1* (the *S. pombe* ortholog of *HMT1*) deletion backgrounds. Similar to our findings for *S. cerevisiae* evolved lines, noise levels of Tdh1-YFP and Gpd3-YFP were significantly increased in *rmt1* mutant cells (Fig 6D). Accordingly, the Hmt1-mediated noise buffering system may represent an important survival strategy that is conserved across diverse microorganisms.

## Discussion

Microorganisms frequently face changing environments. Despite being equipped with complex stress-adaption systems, unpredicted acute stress remains a challenge for cells. Cell populations harboring heterogeneous physiological states enhance their likelihood of surviving environmental fluctuations [56,57]. Many examples of bet-hedging have been reported previously, indicating that this is a common survival strategy among microbes [5, 58]. However, it is not known whether cells exhibit another layer of regulation that allows them to adjust their levels of noise. Our current study provides evidence that Hmt1 can function as a core regulator to constrain or facilitate phenotypic heterogeneity in response to environmental stimuli.

The evolutionary advantage of an environment-sensing noise regulator is readily conceivable. Although stochastic noise is inevitable among individuals within a population, excessive deviation from "normal" levels may result in a significant reduction of fitness under normal conditions. By regulating multiple pathways, Hmt1 allows cells to constrain noise levels. However, when a population is exposed to mild stresses, Hmt1 expression is immediately downregulated, and the noise buffering system is curtailed. The resulting enhanced heterogeneity

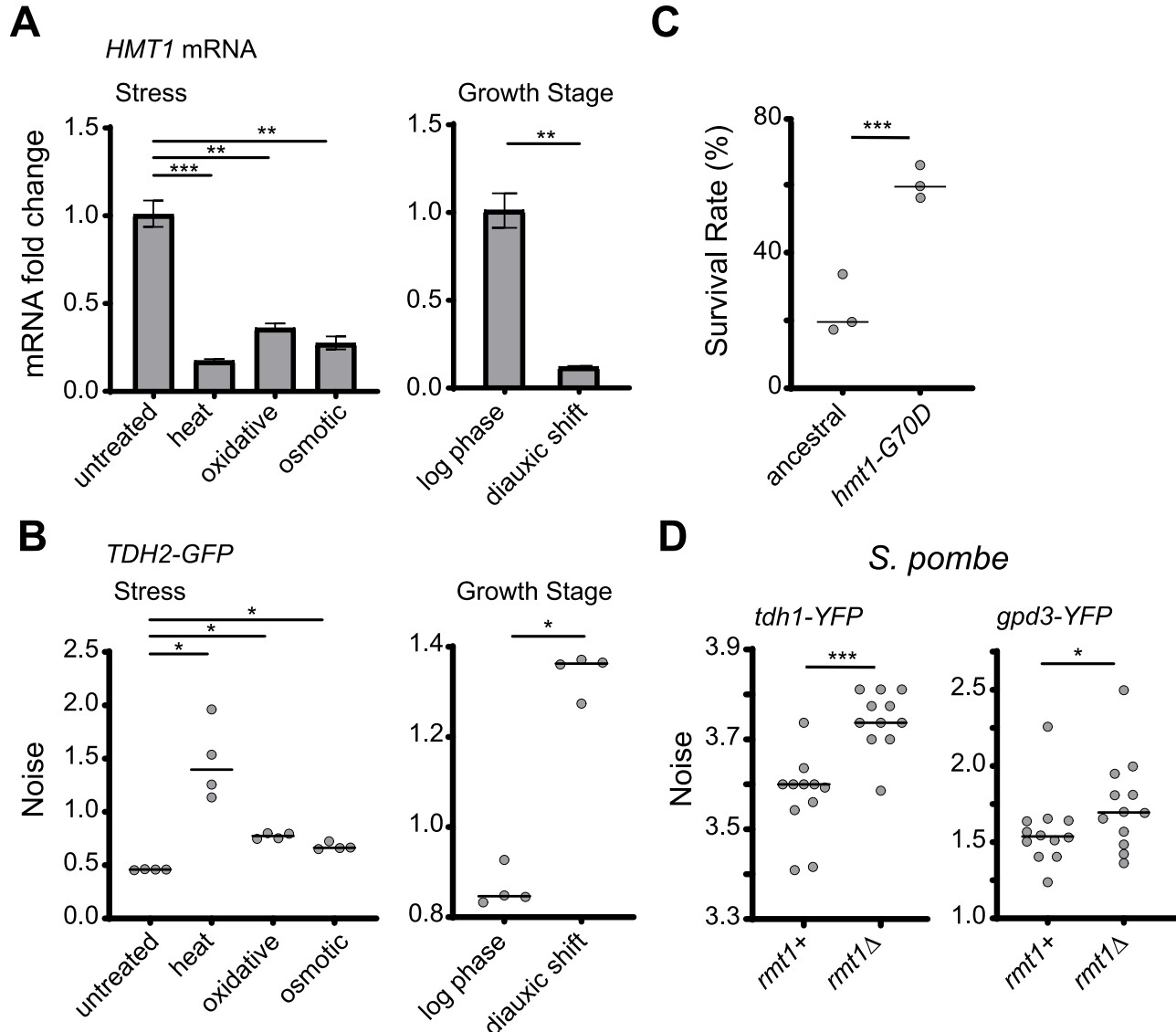

**Fig 6. Hmt1-mediated noise suppression can be released under nonoptimal growth conditions as a conserved cell survival strategy.** (A) *HMT1* transcripts are down-regulated under nonoptimal growth conditions (one-sided paired Student's *t* test, *n* = 3–4; *p* = 0.0008 for heat stress, *p* = 0.003 for oxidative stress, *p* = 0.002 for osmotic stress, *p* = 0.007 for diauxic shift). The level of mRNA was measured using Q-PCR, and *DET1* mRNA was used as the internal control. Error bars indicate standard errors. (B) Nonoptimal growth conditions result in increased Tdh2-GFP noise (one-sided Wilcoxon rank-sum test, *n* = 4; *p* = 0.015 for heat and oxidative and osmotic stress, *p* = 0.014 for diauxic shift). Noise was measured 2.5 h after cells had been shifted to the indicated conditions or was measured at different growth stages. (C) *hmt1-G70D* mutant populations survive better than the ancestral line in medium containing $H_2O_2$ (Fisher's exact test; *p* < 2.2 × $10^{-16}$). Stationary-phase cells were spread on plates with 5.3 mM $H_2O_2$, and survival rates were determined by counting colony-forming units after 5 d of growth. (D) Mutation of the *HMT1* ortholog in the fission yeast *Schizosaccharomyces pombe* also results in increased expression noise. *rmt1* is the ortholog of *HMT1*, whereas *tdh1* and *gpd3* are *TDH2* orthologs. Deletion of *rmt1* increased protein noise of both tdh1-YFP and gpd3-YFP (one-sided Wilcoxon rank-sum test, *n* = 10–12; *p* = 0.0005 for tdh1-YFP, *p* = 0.037 for gpd3-YFP). The median value of replicates is indicated with horizontal solid lines among groups of data points. *p* < 0.05, **p* < 0.01, ***p* < 0.001. Data associated with this figure can be found in the supplemental data file (S1 Data). GFP, green fluorescent protein; Q-PCR, quantitative PCR; YFP, yellow fluorescent protein.

may allow the population to increase its likelihood of survival, especially if the stress is prolonged or escalates (Fig 7).

Hmt1 is a type I arginine methyltransferase that catalyzes monomethylation and asymmetric dimethylation in budding yeast. Hmt1-mediated methylation has been shown to influence

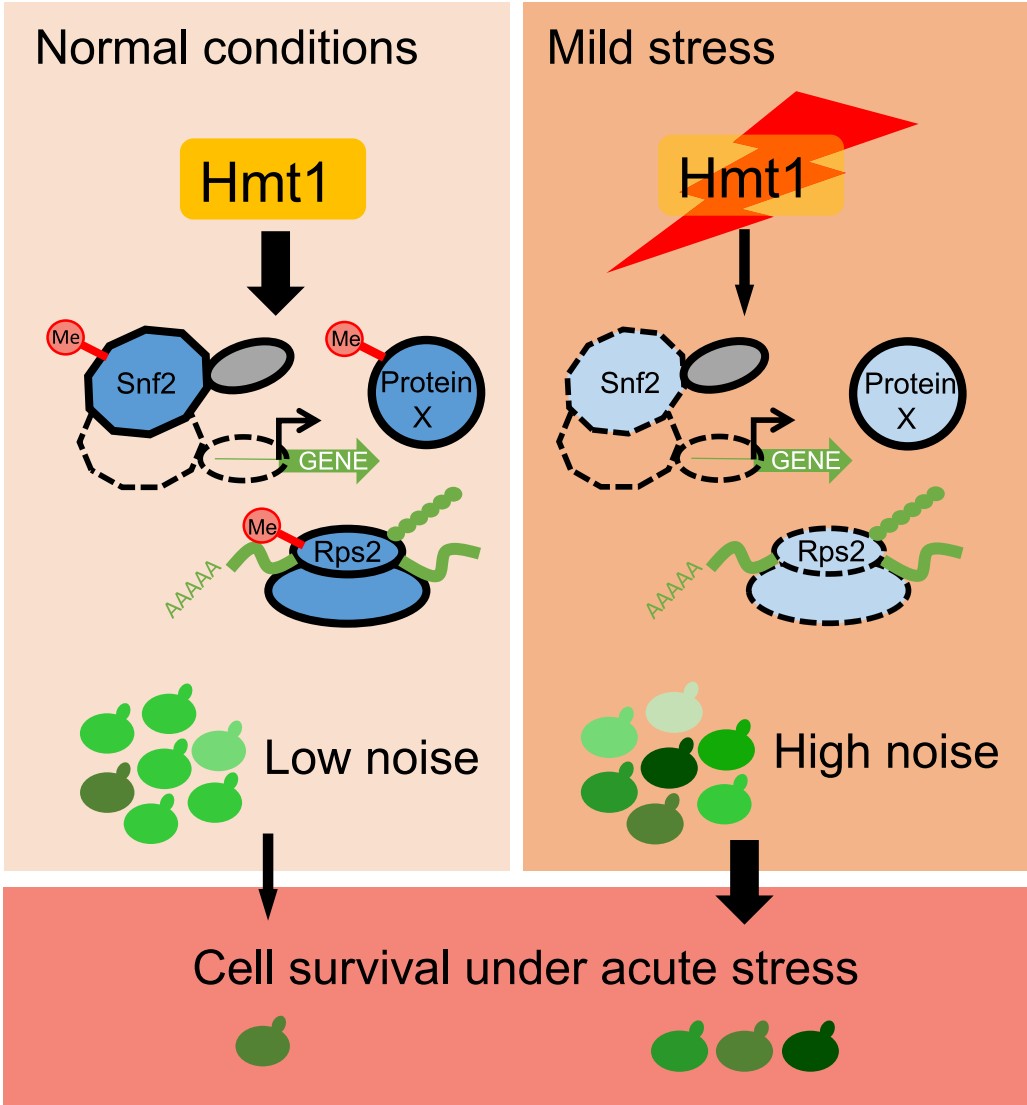

**Fig 7. A model showing how Hmt1 modulates cell-to-cell heterogeneity in response to environmental stress.** Hmt1 methylates and enhances the function of the SWI/SNF chromatin remodeler and small ribosomal subunits to reduce stochastic noise in gene expression. Under normal conditions, cells maintain a high level of Hmt1 and exhibit homogeneous gene expression in most cells. However, when the population encounters environmental stress, *HMT1* expression is down-regulated, inhibiting the functions of Hmt1 targets. Accordingly, expression of Hmt1 gene targets becomes noisier, so individual cells exhibit heterogeneous cell physiologies. The likelihood of population survival is enhanced because of this heterogeneity. Me, methylation.

various cellular pathways, including transcription, posttranscription, and translation [39–41,43,45–49]. Moreover, Hmt1 is a highly interactive protein, ranking among the top 1.5% of the entire budding yeast proteome in terms of the interaction number [59], suggesting it has a central role in coordinating a complex network. Previous studies have shown that another network hub, heat shock protein 90 (Hsp90), can regulate general cellular noise [20,60]. Consistent with predictions from network analyses and results from a genome-wide screen, hub genes have a strong impact on buffering nongenetic and genetic variation [61–63]. More interestingly, both Hsp90 and Hmt1 can work as environmental sensors to adjust noise levels in

response to environmental cues, acting as direct links between noise regulation and adaptive benefits.

Our results show that at least two downstream pathways, i.e., chromatin remodeling and the translational machinery, are involved in Hmt1-mediated noise buffering. The effect of the SWI/SNF chromatin-remodeling complex on gene noise has been reported previously [12]. Here, we have identified Hmt1 as an upstream coordinator of noise regulation and establish the novel role of protein methylation in this process. The detailed mechanism underlying how the ribosomal component Rps2 regulates noise is less clear. Rps2 plays a crucial role in controlling translational accuracy, and its efficiency can be further modulated by posttranslational modifications [64,65]. It is possible that Rps2-mediated noise regulation occurs at the protein translation level. More experiments are required to address this issue.

A recent study has shown that gene expression noise can influence drug resistance and the selection of mutations even in the cells derived from a multicellular organism [66]. The regulatory functions of Hmt1 are generally well conserved in human cells, but they are executed by six orthologs [36,67]. Human orthologs also exist for many Hmt1 substrates, which interact respectively with different human type I enzymes [37,68–70]. For example, the human Hmt1 ortholog protein arginine N-methyltransferase 4 (PRMT4) interacts with the Snf2 ortholog Brahma/SWI2-related gene 1 (BRG1) to facilitate the ATPase activity of the entire human SWI/SNF complex [70]. It is likely that the interactions between Hmt1 and its substrates represent an ancient regulatory network. In support of this supposition, we observed increased noise in *S. pombe* and *S. cerevisiae hmt1* mutants, suggesting that Hmt1-mediated buffering evolved hundreds of millions of years ago, before the ancestors of these two species diverged. Hmt1-mediated noise buffering may also help populations of unicellular *S. pombe* to survive in stressful environments. However, with regard to multicellular organisms, methylation-regulated noise buffering may influence their developmental plasticity, and this interesting topic awaits further study.

## Materials and methods

### Yeast strains and genetic procedures

All *S. cerevisiae* strains used in this study were derived from the W303 strain (*leu2-3, 112 trp1-1 can1-100 ura3-1 ade2-1 his3-11, 15*), and the *S. pombe* strains were derived from the 972 h⁻ strain. Unless otherwise indicated, gene deletion or insertion was based on homologous recombination. Yeast cells were transformed using the lithium acetate method [71] or electroporation under 1,800 V/200 Ω/25 μF (BTX Gemini SC$^2$, Fisher Scientific, Pittsburgh, PA, United States) [72].

All fluorescent fusion proteins were constructed by chromosomal in-frame insertion of the fluorescent protein tag at the 3′ end of the coding region of target genes. The GFP tags were directly amplified from the genomic DNA of corresponding strains in the yeast GFP collection [73]. The BFP tag was amplified from the plasmid pFA6a-link-yomTagBFP2-Kan [74]. Prior to constructing *ADK1-GFP* and *TDH2* promoter-driven *GFP* in ancestral and evolved *TDH2-GFP*-carrying lines, the *GFP* of *TDH2-GFP* in these strains was removed and replaced with a stop codon and the *TDH2* 3′ untranscribed region (UTR). For *TDH2* promoter-driven *GFP*, 1,000 base pairs (bp) upstream of the *TDH2* coding region was fused with *GFP*, and the fused fragment was inserted between positions 204886 and 204887 of chromosome I [75]. For YFP tagging in *S. pombe*, the coding region of yVenus [76] without the start codon was fused with hphMX6 by two-fragment PCR. To delete *S. cerevisiae* genes, KanMX4-containing DNA fragments for homologous recombination were amplified from the genomic DNA of corresponding strains in the yeast deletion collection [77]. To construct a hypomorphic mutant of

*RPS2*, we directly amplified the *rps2*-DAmP allele from the yeast DAmP diploid collection [78] and used it to replace the native gene. To delete *rmt1* in *S. pombe*, a KanMX6-containing DNA fragment with homologous flanking regions was generated and used to replace the whole gene.

All single nucleotide polymorphism (SNP) mutant reconstitution strains were constructed using the CRISPR/Cas9 system [79,80]. Briefly, host cells were transformed with the Cas9 expression plasmid, and then the transformant was introduced with a DNA fragment encoding gRNA, a linearized vector, and mutation-containing donor DNA fragments. After transformation, single colonies were streaked out on 1% yeast extract, 2% peptone, and 2% dextrose (YPD) plates to purify the CRISPR transformants. Genomic DNA of the resulting single colonies was isolated and examined initially by allele-specific PCR [81] and then by Sanger sequencing. The plasmids for Cas9 and gRNA expression are listed in S3 Table.

## Experimental evolution

For the evolution experiment, eight genes (*ADK1*, *APA1*, *PCM1*, *RPL4B*, *SAM4*, *TDH2*, *TPD3*, and *TYS1*) were selected and fused with GFP to generate evolving strains, each carrying an individual reporter gene. These reporter genes were chosen based on the following criteria: (1) each gene belongs to a distinct cellular pathway, (2) the expression level of GFP fusion proteins is constant during the cell cycle and is sufficiently high to be detectable by flow cytometry, and (3) the reporter proteins are evenly localized in the nucleus or cytosol to avoid misinterpretation due to dynamic organelles.

Because of the complexity of our selection regime, only one evolving line for each reporter gene was established. In each selection cycle, $1 \times 10^6$ cells were treated with 2.8% EMS (see the EMS mutagenesis section below for details) to increase the genetic diversity of the cell population. This mutagenic treatment is crucial, as we initially ran a pilot experiment using a similar selection regime but lacking the mutagen treatment and did not observe any obvious increase in noise after 70 cycles of selection. However, our EMS treatments also significantly increased the number of mitochondria-defective cells, which are known to increase population heterogeneity. To constrain the population of mitochondria-defective cells, we grew cells in nonfermentable 1% yeast extract, 2% peptone, and 2% glycerol (YPG) medium at 28 ˚C after EMS treatments. After 12 h of growth in YPG, cells were sorted to select 5,000 cells from the top (or bottom) 5% of the total population in terms of GFP intensity (see the Fluorescence-activated cell sorting section below for details). These cells were grown in 3 ml YPG for approximately 36 h to reach $OD_{600} = 1$ before proceeding to the next cycle of selection. We used alternating selection between the top 5% and bottom 5% of total populations throughout the evolution experiment. The effective population size was estimated to be $1.33 \times 10^5$ cells using the formula $2/N_e = 1/(N_{01} \times g) + 1/(N_{02} \times g)$, in which $N_0$ is the initial population size and $g$ is the number of generations during each growth period [82].

After 35 cycles of selection, five individual clones were isolated from each evolved population, and their GFP noise levels were measured. The clone with the greatest increase in noise without exhibiting a decrease in the mean intensity of GFP or increased variation in cell size was selected for further genetic analysis.

## EMS mutagenesis

Mutagenesis was performed according to a previously published protocol by which mutation rates can be increased without inducing considerable cell death [83]. Briefly, we washed $1 \times 10^6$ cells with sterile water once and with 100 μl of phosphate buffer (0.1 M $Na_2HPO_4$ [pH 7.0]) once and then resuspended them in 90 μl of phosphate buffer. We added 90 μl of EMS-

containing phosphate buffer to the cell solution, resulting in a final concentration of 2.8% EMS. The solution was maintained under constant shaking at room temperature for 30 min, before stopping the reaction by adding 50 μl of 25% sodium thiosulfate. After washing with sterile water, the cells were transferred to 10 ml YPG. The survival rate of wild-type cells after EMS treatment ranged from 60% to 80%.

### Fluorescence-activated cell sorting and flow cytometry

Cells were suspended in filtered phosphate-buffered saline (PBS) (137 mM NaCl, 2.7 mM KCl, 10 mM $Na_2HPO_4$, and 1.8 mM $KH_2PO_4$) and sampled using a BD FACSJazz machine (BD Bioscience, Franklin Lakes, NJ, USA) at an event rate of ≤3,000 cells/s. GFP readouts were acquired with bandpass filters of 513/17 nm using laser excitation at 488 nm. For YFP, we used laser excitation at 488 nm with a 542/27-nm filter, and for BFP, we used a 450/50-nm filter and laser excitation at 405 nm.

To eliminate interference from small particles in the fluidic system, we excluded particles with forward scatter (FSC) signals < 2,570 (as a trigger threshold). We established three hierarchical gates. First, to avoid cell aggregates, we collected cells that had relatively constant signals of trigger pulse width along FSC signals. Second, to eliminate possible cell aggregates, we collected cells that had relatively constant signals of FSC-W along FSC signals. Third, to avoid cell debris and abnormally large cells, we collected cells having FSC and side scatter (SSC) signals both ranking within 5%–95% of the population.

For alternating selection in experimental evolution, we applied the trigger threshold and the first and third gates. Based on the distribution of GFP levels, we defined the highest or lowest 5% of the total gated population and collected 5,000 cells. The collected cells were then grown in 3 ml YPG and used for the next run of selection. For fitness assays, we applied the trigger threshold and all three gates. Based on the distribution of GFP levels, we defined the high and low subpopulations from the total gated population and collected individual cells. A 1.0 single-drop sorting mode was employed. The viability of the collected cells was then measured under different conditions.

### Noise and signal distribution measurement

We measured the noise of fluorescent protein signals by the Fano factor (a ratio of variance to the mean, %) using more than 5,000 early log-phase cells. This measurement is characterized by having less interference from the mean and is more sensitive to variation-driven increased noise [12,84]. We used the same trigger threshold and the first two hierarchical gates as employed for cell sorting in our noise measurement. For the third gate, we used contour plots of FSC and SSC signals to collect cells constituting 60% of the second-gated population to ensure more homogenous cell size and cell physiology. Fluorescence readouts of the entire population were log-transformed and used to calculate the noise.

### Fitness assays under different growth conditions

To measure $H_2O_2$ resistance, Tdh2-GFP-carrying cells were grown in YPD for 5 d to ensure that the cells had entered stationary phase. More than 100 cells from the top 30% or bottom 30% of the gated population in terms of their Tdh2-GFP intensity were sorted and spotted onto plates with or without 4.4 mM $H_2O_2$. The survival rates were measured after 5 d of incubation at 28 °C.

For heat-resistance assays, cells from the top 10% or bottom 10% in Tdh2-GFP intensity were sorted into microcentrifuge tubes. The collected cells were divided into two parts. One part was immediately placed in a PCR machine to perform heat ramping from 30 °C to 56 °C

for 20 min as a heat stress [85]. The other part was placed in fresh YPD media for 2.5 h and then subjected to the same heat treatment. Cell survival rates were measured as colony-forming units after 3 d of incubation at 28 ˚C.

As a rebudding assay, more than 100 unsorted stationary-phase cells were placed on agarose pads and monitored by time-lapse microscopy at 15-min intervals for at least 12 h [86]. Only unbudded cells at the beginning of the recording period were monitored.

### Western blotting

Log-phase cells were lysed using NaOH lysis [87]. Proteins in the lysates were separated with SDS-PAGE and transferred to a PVDF membrane. Mouse anti-GFP antibody (1:4,000) (#sc-9996, Santa Cruz Biotechnology, Dallas, TX, USA) was used to detect Tdh2-GFP. Rabbit anti-G6PDH antibody (1:4,000) (#A9521, Sigma-Aldrich, St. Louis, MO, USA) was used to detect G6PDH, which served as an internal control. Mouse anti–methylated arginine antibody (mab0002-P, Covalab, Villeurbanne, France) was used to detect proteins with methylated arginine. MultiMab rabbit monoclonal mix antibodies against an asymmetric di-methyl arginine motif (#13522, Cell Signaling Technology, Danvers, MA, USA) were used to detect proteins with asymmetric di-methylated arginine.

### Microscopy

Microscopy was conducted using a 60× objective lens and an ImageXpress Micro XL system (Molecular Device, Sunnyvale, CA, USA).

### F1 segregant analysis

The evolved line was mated with the ancestral line, and the resultant diploid cells were induced to sporulate. Sporulated culture was harvested into a microcentrifuge tube and treated with 0.5 mg/ml Zymolyase-100T (Nacalai Tesque, Kyoto, Japan) in 1 M sorbitol at 28 ˚C for 2 h to remove the ascal wall. Cells were then treated with 2% SDS at 28 ˚C for 10 min to kill unsporulated diploid cells, before washing with sterile water and vortexing vigorously to attach individual spores to the tube wall. We then added 0.01% Triton X-100 solution to the tube, before vigorous sonication to detach spores from the tube wall and to separate spore clusters. The suspension was diluted and spread on YPD plates to isolate F1 haploid segregants.

We conducted a total of three runs of noise measurement to identify "evolved-like" and "ancestral-like" F1 segregants. The progenies ranking within the top 20% or bottom 20% of the Tdh2-GFP noise level without changing the mean intensity (i.e., within three standard deviations of the control) were selected for the next run of noise measurement. We started with 360 F1 progeny of confirmed ploidy in the first round and obtained 16 "evolved-like" segregants and 20 "ancestral-like" segregants after finishing the third run. These cells were subjected to whole-genome sequencing analysis.

### Whole-genome sequencing analysis

Yeast cells were resuspended in 200 μl of lysis buffer (2% Triton X-100, 1% SDS, 100 mM NaCl, 10 mM Tris-HCl [pH 8.0], and 1 mM Na₂EDTA). Glass beads (0.5 mm in diameter) that amounted to the volume of the cell pellet were added, followed by the addition of 200 μl of phenol:chloroform:isoamyl alcohol (PCIA, 25:24:1 [pH 8.0]). The mixture was vortexed for 5 min. We then added 200 μl of TE buffer (10 mM Tris·Cl; 1 mM Na₂EDTA [pH 8.0]), and the mixture was centrifuged for 5 min at maximum speed. Only 350 μl of the solution in the aqueous layer was extracted and subjected to EtOH precipitation. The pellet was air-dried and then

dissolved in 400 μl of TE buffer containing 75 μg/ml RNase A. The mixture was then incubated at 37 ˚C for 5 min. We then added 200 μl of PCIA, and we inverted the tubes to mix. EtOH precipitation was then performed at room temperature under constant shaking for 2 h. The pellet was air-dried and then dissolved in 100 μl of sterile double-distilled water. The ratio of $OD_{260}$ to $OD_{280}$ of the purified DNA ranged from 1.8 to 2.1. Concentrations of genomic DNA were determined using a Qubit dsDNA BR assay kit (ThermoFisher Scientific).

Equal amounts of genomic DNA from individual "evolved-like" or "ancestral-like" F1 segregants were pooled. These two DNA pools were sequenced using an Illumina NextSeq500 system (Illumina, San Diego, CA, USA) with 150-bp paired-end reads from 350-bp libraries. At least 57× coverage was achieved for sequencing ancestral and evolved clones, and we obtained 150–200× coverage for the pooled segregants. Sequence results were analyzed using CLC Genomics Workbench 9.1 with default settings for read import, read trimming, read alignment, duplicate-read removal, local realignment, SNP calling (with the slight modification of using a 10% frequency cut-off in the "ancestral-like" pool to retrieve as many mutations as possible, since the default is 35%), and SNP annotation. We used the reference genome of S288C (version R64-2-1) from The *Saccharomyces* Genome Database for SNP calling because it is the best-annotated.

## Identifying candidate causal SNPs responsible for increased noise

By using the CLC Genomics Workbench 9.1 function "filter by control read" with a criterion of fewer than 2 control reads, we could eliminate SNPs existing in both ancestral and evolved lines (S1 Table). The resulting list of evolved SNPs was then cross-referenced with the list of SNPs from the "evolved-like" pool. We anticipated that candidate causal SNPs responsible for increased noise would be enriched in the "evolved-like" but not in the "ancestral-like" pools. By simulating sequencing results (using Python 2.7) from 150× coverage, we determined the frequencies of SNPs that were randomly segregated into two pools of 10 or 20 individuals to represent SNP enrichment thresholds for increased noise [88]. We identified SNPs with a frequency > 70% in the "evolved-like" pool and with a frequency < 32% in the "ancestral-like" pool at a 90% confidence interval as being correlated with increased noise (S2 Table). The thresholds are represented by 90% confidence intervals for simulations with 10 individuals and at 95% for those with 20 individuals. The script used to determine the threshold of allele frequency for the causal mutations was uploaded to GitHub under a repository "imst0715" and can be found by searching "Threshold-for-bulk-segregant-analysis."

## Histone ChIP combined with Q-PCR

Log-phase cells (approximately 30 $OD_{600}$) were harvested and cross-linked by 1% formaldehyde at 30 ˚C for 30 min. After quenching with 125 mM glycine, the cell suspension was lysed by three cycles of 5 min beating and 1 min cooling at 4 ˚C of 0.5-mm glass beads in FA buffer (50 mM HEPES-KOH [pH 7.5], 140 mM NaCl, 1% Triton X-100, 0.1% SDS, and 0.1% Deoxycholate Na salt) supplemented with 1x protease inhibitor cocktail (Protease Inhibitor Cocktail Set IV in DMSO; Merck, 539136) and 1 mM PMSF. We made a hole in the bottom of the lysate-containing tube to allow the lysate to flow through into a collection tube upon centrifugation at 500*g* for 3 min at 4 ˚C. The chromatin fraction was pelleted down by centrifugation at 12,000*g* for 15 min at 4 ˚C and washed twice with FA buffer. We transferred all of the chromatin suspension (2 ml) into a 15-ml centrifugation tube (BIOFIL), avoiding bubbles. We sheared the chromatin using a Bioruptor (Diagenode, Denville, NJ, USA), with 5 cycles of 15 min (30 s on and 30 s off per minute) at "High intensity" mode at 4 ˚C. Ice-cold water was resupplied in the Bioruptor tank after each cycle to maintain the low temperature. The sheared

chromatin was cleared by centrifugation at 12,000*g* for 15 min at 4 ˚C. One-fifteenth of the cleared chromatin was used as input and stored at −80 ˚C. The remaining cleared chromatin was transferred into a tube containing preincubated Dynabeads Protein A (#10002D, Invitrogen, Waltham, MA, USA) with histone 3 antibody (ab#1791, Abcam), followed by end-over-end mixing overnight at 4 ˚C. The beads were anchored with DynaMag-2 Magnet (Thermo-Fisher Scientific, Waltham, MA, USA) to remove unbound molecules and were then sequentially washed with 1 ml of FA buffer, 500 mM NaCl-containing FA buffer, DOC buffer (10 mM Tris-HCl, 1 mM $Na_2EDTA$ [pH 8.0], 250 mM LiCl, 0.5% NP-40, 0.5% Deoxycholate Na salt), and TE buffer (10 mM Tris·Cl; 1 mM $Na_2EDTA$ [pH 8.0]). The bound molecules were eluted by adding TES buffer (10 mM Tris·Cl; 1 mM $Na_2EDTA$ [pH 8.0], and 1% SDS) and incubating at 65 ˚C for 20 min, followed by a second elution with TE buffer (65 ˚C for 10 min). The two eluents were combined. The eluent and thawed input sample were incubated in 0.125 µg/ml RNase A at 37 ˚C for 30 min, followed by Proteinase K treatment (2 mg/ml Proteinase K at 42 ˚C for 1 h). The samples were de-cross-linked at 65 ˚C overnight. DNA was purified using a QIAquick DNA Purification Kit (Qiagen, Hilden, Germany) according to the manufacturer's instructions, with the modification of a two-cycle wash step using Qiagen PE buffer. DNA (0.25 ng) was then subjected to Q-PCR with the primers for nucleosome-occupied and nucleosome-depleted regions of the promoter and coding regions for *TDH2* and *GLY1* (S3 Table) [89]. The amplicons for *TDH2* and *GLY1* regions were normalized to the amplicon of a well-defined nucleosome-occupied region at *GAL1* promoter [90]. The nucleosome occupancy was determined by a ratio of the normalized amplicons from the ChIP samples to those from the input samples.

### Q-PCR of mRNA under different growth conditions

For stress conditions, log-phase cells grown in YPD media were divided into four parts. One part was continuously grown in YPD as an untreated sample, and the remaining three parts were treated with 0.375 M KCl for 20 min or 0.4 mM $H_2O_2$ for 20 min or subjected to 42 ˚C for 30 min. For different growth states, total RNA was extracted from the same batch of cultures when cells were in log phase and entering diauxic shift. To investigate the activity of SWI/SNF chromatin-remodeling complexes, cells were first grown to log phase in Serine-depleted complete synthetic mixture (CSM-Serine). Then, L-Serine was added to a concentration of 0.1% to induce *CHA1* expression, before harvesting cultures 0, 5, and 15 min after induction, with the addition of 0.05% $NaN_3$ to stop transcription [52].

RNA from 5–10 $OD_{600}$ of cells was extracted and quantified according to a previous report [72] with modifications. Cells were harvested at 4 ˚C. RNA quality was examined using an Agilent RNA 6000 Nano kit with an Agilent 2100 Bioanalyzer (Agilent Technologies). RNA (2 µg) was reverse-transcribed with 0.5 µg oligo(dT)$_{18}$ primer using an Applied Biosystem High-Capacity cDNA Reverse Transcription kit (ThermoFisher Scientific). The reaction products (1 µl) were then used for Q-PCR in an Applied Biosystem 7500 Fast Real-Time PCR system (ThermoFisher Scientific) with gene-specific primers (S3 Table).

### Statistical analyses

All statistics were performed using R language (http://www.r-project.org/).

### Supporting information

**S1 Fig. Sorted subpopulations of cells exhibiting low or high Tdh2-GFP are probably not genetically distinct.** Cells were sorted according to their Tdh2-GFP levels into low (red circle) and high (blue circle) subpopulations, each of which constituted 10% of the whole population

(gray). These cells were grown in YPD for the indicated periods of time and analyzed by flow cytometry. The distributions of the low (red) and high (blue) Tdh2-GFP subpopulations are superimposed with that of the parental population (gray outline). The vertical dashed lines indicate the mean signal intensity of the parental population. Data associated with this figure can be found in the supplemental data file (S1 Data). GFP, green fluorescent protein; YPD, 1% yeast extract, 2% peptone, and 2% dextrose.
(EPS)

**S2 Fig. Experimental evolution alters the Tdh2-GFP signal distribution without affecting protein integrity or subcellular localization.** (A) Increased noise in the evolved Tdh2-GFP-carrying line is due to a flattened distribution rather than a bimodal one. Data associated with this figure can be found in the supplemental data file (S1 Data). (B) Full-length Tdh2-GFP is retained after evolution. Western blots were hybridized using mouse anti–GFP antibody (1:4,000) (the upper blot) and rabbit anti–G6PDH antibody (1:4,000) (the lower blot). The original image of the blot can be found in the supplemental file (S1 Raw Images). (C) Protein localization of Tdh2-GFP is not altered by our experimental evolution approach. Cells were imaged using a 60× objective under the FITC channel (the upper panel) or bright field (the lower panel). The scale bar is 5 μm. FITC, fluorescein isothiocyanate; G6PDH, glucose-6-phosphate dehydrogenase; GFP, green fluorescent protein.
(EPS)

**S3 Fig. Tdh2-GFP signal in F1 progeny selected for whole-genome sequencing.** For bulk segregant analysis, 360 F1 progeny were derived by backcrossing the evolved clone to the ancestral clone. The noise and mean of Tdh2-GFP signal in individual progeny were analyzed, and a total of three runs of noise measurement were conducted to identify "evolved-like" and "ancestral-like" F1 segregants. Data from the final run of analysis are shown here. Genomic DNA of the "evolved-like" and "ancestral-like" F1 progeny was extracted and respectively pooled for whole-genome sequencing. Data associated with this figure can be found in the supplemental data file (S1 Data). GFP, green fluorescent protein.
(EPS)

**S4 Fig. The *hmt1-G70D* mutation phenocopies the loss-of-function mutation.** (A) The identified G70D mutation of Hmt1 is located in a highly conserved methyltransferase motif. An alignment of the primary sequences of the conserved motif is shown for various methyltransferases from budding yeast, fission yeast, human, and bacteria. Residues shared with *S. cerevisiae* Hmt1 are labeled in yellow, and the mutated glycine residue observed in the evolved clone is indicated by an arrowhead. (B) Both *hmt1-G70D* and Hmt1 deletion mutants exhibit a similar level of increased Tdh2-GFP noise (one-sided Wilcoxon rank-sum test, $n$ = 5–10; $p$ = 0.0013 for *hmt1-G70D*, $p$ = 0.0013 for *hmt1Δ*). Data associated with this figure can be found in the supplemental data file (S1 Data). (C) The G70D mutation of Hmt1 results in defective methylation. The patterns of asymmetric dimethylation (left) and panmethylation (right) of arginine in whole-cell lysates of *hmt1-G70D* mutant cells did not differ from those of deletion mutants (*hmt1Δ*) but differed from those of wild-type cells. The arrowheads indicate the signals differentially detected between methylation-competent and methylation-deficient strains. The original image of the blot can be found in the supplemental file (S1 Raw Images). GFP, green fluorescent protein.
(EPS)

**S5 Fig. Tdh2-GFP noise significantly increases shortly after stress treatments.** Tdh2-GFP noise increases under nonoptimal growth conditions (one-sided Wilcoxon rank-sum test, $n$ = 4; $p$ = 0.015 for heat stress, $p$ = 0.015 for oxidative stress). Noise was measured after cells

were treated with the indicated stress for 20–30 min. The difference between untreated and treated cells became more obvious after 2 h (Fig 6B), which probably reflects the time it takes for cells to alter the abundance of Tdh2-GFP protein. The median value of replicates is indicated with horizontal solid lines among groups of data points. *$p < 0.05$. Data associated with this figure can be found in the supplemental data file (S1 Data). GFP, green fluorescent protein.
(EPS)

**S1 Table. Mutations in the evolved *TDH2-GFP*-carrying strain.** *GFP*, green fluorescent protein.
(XLSX)

**S2 Table. Identities and frequencies of candidate mutations responsible for increased noise, and identified mutations in the ancestral-like and evolved-like F1 progeny pools.**
(XLSX)

**S3 Table. Plasmids and primers used in this study.**
(XLSX)

**S1 Data. Data files related to Figs 1B, 2A–2C, 3A, 3B, 4A–4C, 5A–5E, and 6A–6D, as well as S1, S2A, S3, S4B, and S5 Figs.**
(XLSX)

**S1 Raw Images. The original images of the blots shown in S2B and S4C Figs.**
(PDF)

## Acknowledgments

We thank members of the Leu lab for helpful discussion and comments on the manuscript. We thank Shao-Win Wang for the *S. pombe* strains. We also thank John O'Brien for manuscript editing and Po-Hsiang Hung and the IMB Bioinformatics, Genomics, and Imaging cores for technical assistance.

## Author Contributions

**Conceptualization:** Jun-Yi Leu.

**Formal analysis:** Shu-Ting You, Jun-Yi Leu.

**Funding acquisition:** Jun-Yi Leu.

**Investigation:** Shu-Ting You, Yu-Ting Jhou, Jun-Yi Leu.

**Methodology:** Cheng-Fu Kao.

**Project administration:** Jun-Yi Leu.

**Resources:** Jun-Yi Leu.

**Supervision:** Jun-Yi Leu.

**Validation:** Shu-Ting You.

**Writing – original draft:** Shu-Ting You, Jun-Yi Leu.

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
