## [Editor Report · Decision Letter 0]

22 Jul 2019

Dear Dr Leu, 

Thank you for submitting your manuscript entitled "Experimental evolution reveals a general role for the methyltransferase Hmt1 in noise buffering" for consideration as a Research Article by PLOS Biology.

Your manuscript has now been evaluated by the PLOS Biology editorial staff as well as by an Academic Editor with relevant expertise and I am writing to let you know that we would like to send your submission out for external peer review.

**Important**: Please also see below for further information regarding completing the MDAR reporting checklist. The checklist can be accessed here: https://plos.io/MDARChecklist

Please re-submit your manuscript and the checklist, within two working days, i.e. by Jul 24 2019 11:59PM.

Kind regards,

Hashi Wijayatilake, PhD,

Managing Editor

PLOS Biology

INFORMATION REGARDING THE REPORTING CHECKLIST:

PLOS Biology is pleased to support the "minimum reporting standards in the life sciences" initiative (https://osf.io/preprints/metaarxiv/9sm4x/). This effort brings together a number of leading journals and reproducibility experts to develop minimum expectations for reporting information about Materials (including data and code), Design, Analysis and Reporting (MDAR) in published papers. We believe broad alignment on these standards will be to the benefit of authors, reviewers, journals and the wider research community and will help drive better practise in publishing reproducible research. 

We are therefore participating in a community pilot involving a small number of life science journals to test the MDAR checklist. The checklist is intended to help authors, reviewers and editors adopt and implement the minimum reporting framework. 

IMPORTANT: We have chosen your manuscript to participate in this trial. The relevant documents can be located here:

MDAR reporting checklist (to be filled in by you): https://plos.io/MDARChecklist

**We strongly encourage you to complete the MDAR reporting checklist and return it to us with your full submission, as described above. We would also be very grateful if you could complete this author survey:

https://forms.gle/seEgCrDtM6GLKFGQA

Additional background information:

Interpreting the MDAR Framework: https://plos.io/MDARFramework

Please note that your completed checklist and survey will be shared with the minimum reporting standards working group. However, the working group will not be provided with access to the manuscript or any other confidential information including author identities, manuscript titles or abstracts. Feedback from this process will be used to consider next steps, which might include revisions to the content of the checklist. Data and materials from this initial trial will be publicly shared in September 2019. Data will only be provided in aggregate form and will not be parsed by individual article or by journal, so as to respect the confidentiality of responses. 

Please treat the checklist and elaboration as confidential as public release is planned for September 2019.

We would be grateful for any feedback you may have.

---

## [Decision Letter · Decision Letter 1]

6 Sep 2019

Dear Dr Leu,

Thank you very much for submitting your manuscript "Experimental evolution reveals a general role for the methyltransferase Hmt1 in noise buffering" for consideration as a Research Article by PLOS Biology. As with all papers reviewed by the journal, yours was evaluated by the PLOS Biology editors as well as by an Academic Editor with relevant expertise and by independent reviewers. I sincerely apologize again for the unusual delay in getting you this decision. We did however encounter delays over the busy summer month of August during which reviewer availability is limited and extensions were needed, and we wanted to ensure thorough review. 

As you can see, there is a lot of enthusiasm for this work. The reviewers appreciated the attention to this topic and the rigorous and thoughtful study design. Based on the reviews, we will likely accept this manuscript for publication, providing that you will modify the manuscript according to the review recommendations. We have discussed Reviewer 1's requests, relating to the exact biochemical mechanisms and the evolutionary implications, with the Academic Editor. While we consider them very helpful, we will not require additional data to address these points as we acknowledge they go beyond the scope of the current study and do not affect the main conclusions. Therefore please address these points by toning down the relevant arguments, especially the putative links between HMT1-TDH2-bet hedging.

We expect to receive your revised manuscript within two weeks. Your revisions should address the specific points made by each reviewer. In addition to the remaining revisions and before we will be able to formally accept your manuscript and consider it "in press", we also need to ensure that your article conforms to our guidelines. A member of our team will be in touch shortly with a set of requests. As we can't proceed until these requirements are met, your swift response will help prevent delays to publication.

******* 

Please note that you may have the opportunity to make the peer review history publicly available. The record will include editor decision letters (with reviews) and your responses to reviewer comments. If eligible, we will contact you to opt in or out.

Early Version

Sincerely,

Hashi Wijayatilake, PhD, 

Managing Editor

PLOS Biology

DATA POLICY:

Thank you for depositing you whole-genome sequence data at NCBI BioProject (PRJNA552720). Please also ensure that you provide the individual numerical values that underlie the summary data displayed in all the following figure panels as they are essential for readers to assess your analysis and to reproduce it:

Figs. 1B, 2A-C, 3AB, 4A-C, 5A-E, 6A-D, 

S1, S2A, S3, S4B, S5

**Please also ensure that each figure legend in your manuscript include information on where the underlying data can be found.

**Please ensure that your Data Statement in the submission system fully describes where your data can be found.

For manuscripts submitted on or after 1st July 2019, we require the original, uncropped and minimally adjusted images supporting all blot and gel results reported in an article's figures or Supporting Information files. We will require these files before a manuscript can be accepted so please prepare them now, if you have not already uploaded them. Please carefully read our guidelines for how to prepare and upload this data: https://journals.plos.org/plosbiology/s/figures#loc-blot-and-gel-reporting-requirements.

REVIEWS:

Reviewer #1: 

The manuscript from You et al. describes a laboratory evolution experiment where population of yeast cells expressing a fluorescently-tagged enzyme were subjected to selection alternating for its high and low expression, respectively. The authors established evolved lines where noise in expression of the reporter protein ('protein noise') is increased as compared to the ancestor strain. They identify a mutation in HMT1 that largely contributes to this effect. They show that this mutation impairs the methyl-transferase activity of the enzyme, that it increases expression noise of various reporters (from related as well as unrelated pathways), and that ablation of HMT1 activity can increase expression noise not only in S. cerevisiae but also in S. pombe. All these demonstrations are nicely presented, with rigorous controls and convincing experimental results and I wish to congratulate the authors for the amount and quality of the work and for these findings.

The authors also address two other points in the manuscript and I see important issues in each one of them.

1. The first point is the mechanism of action of HMT1 on noise. Given the large spectrum of protein substrates that HMT1 modifies, this is a difficult question. The authors tested the effect of mutating four known targets (null or hypomorphic alleles), and they found an increase in noise for two mutants (rps2 and snf2), a decrease for one mutant (npl3) and no effect for the fourth mutant (sbp1). They tested the activation of CHA1, for which Swi/Snf is needed, in a hmt1-G70D mutant and found that is was deficient. They also report that the noise increase of Tdh2::GFP and Gly1::GFP is accompanied by an increase of nucleosome occupancy at their promoters in the hmt1-G70D mutant. They conclude that Rps2 and SWI/SNF complex are relevant targets of HMT1 in controlling noise.

The data is of course consistent with this, but it is not a demonstration. For this claim (sentence 'is primarily achieved via two Hmt1 methylation targets' in abstract) to be demonstrated, the authors should perform double-mutant analysis: if the hmt1-G70D does not further increase noise in a snf mutant, and does not further reduce CHA1 activation in a snf mutant, then yes, SNF is needed for the effect of HMT1. If it does, then there are other targets that mediate the effect of HMT1. 

Also, in all these analyses, the mean expression level of the reporter must be presented. Was it affected in some of the mutants?

2. The second point is whether HMT1 relays environmental sensing to expression noise, which would make it a fitness bet-hedging actor. The authors reach this conclusion on the basis of i) expression of HMT1 is reduced under stress (6A) ii) hmt1-G70D cells have a higher survival rate under stress (6C) iii) TDH2-GFP noise is higher under stress (6B) and iv) stationary TDH2-GFP-high cells better survive stress and re-germinate faster when re-fed (Fig 2). I agree that TDH2-GFP expression heterogeneity correlates with survival/fitness differences, but the data does not demonstrate that this happens via HMT1. If a change in HMT1 expression in response to stress modifies TDH2-GFP noise, then:

- forcing HMT1 expression to be constant (with a constitive promoter) despite stress should eliminate the effect of stress on TDH2-GFP noise and, possibly, decrease survival rate upon stress.

- TDH2-GFP noise should not increase (or not as much) upon stress in the hmt1-G70D mutant. The authors could simply repeat the experiment of Fig 6B and S5 on hmt1-G70D.

Without this, the action of HMT1 on noise is demonstrated but its implication on bet-hedging is not.

Other less-important recommendations:

- The term "protein noise" is used throughout the article and must be rigorously defined in text.

- Did the authors test if a drug inhibitor of HMT1, such as AMI-1, can increase noise ? This would be an important result because such inhibitors are considered for the clinics, and a widespread effect on expression noise may have important consequences, especially for cancer. See Hü and J Curr. Op. in Chem. Biol. 2017 for a review.

- Fig2B: I see a linear decrease in the [10,17] mean GFP range and then a flat line at mean > 17. There could be a threshold at about ~17 where 'protection' is maximized.

- As written in discussion, the strong effect of snf6 deletion on noise was reported long ago (ref 11). This must also be cited in the results section (page 12 lines 9-11).

- Fig7 tends to reduce the mechanistic model to Snf2 and Rps2 intermediate actors, while there are probably many others. This can mislead readers to focus their attention on these two HMT1 substrates only.

- Fig S4C: meaning of arrowheads ?

-page 4, line 18: 'regularly face changing env' instead of 'constantly face changing env'?

- page7, line23: 'some populations', not 'same populations'.

- Fig6D: writing S. pombe or 'fission yeast' on the figure as a header would help.

--

Reviewer #2: 

In this paper You and Leu performed a yeast experimental evolution to find genes that regulate noise in gene expression. They started with a clever (though not very intuitive) strategy of repeated cycles of selection for cells that express tagged proteins highly and then lowly. After 35 such cycles they were able to increase level, without a trivial reduction in mean expression for two of the proteins. They speculated that they have achieved evolution for higher noise. They verified that the enhanced noise in the evolved strain was not specific to the gene that was used in evolution, indicating reassuringly that they have probably affected a general mechanism. They went on to sequence the genome and found a very high number of mutations (due to mutagen indeed) , but they followed up with a clever segregant analysis to scale down the number of candidates to a handful. They then found a hit, hmt1, and a single amino acid mutation in it, that appeared sufficient for noise enhancement (by putting the mutation in the wild-type, and by reverting it in the evolved strain). This methy-transferase appears to regulate noise through methylatin of diverse transcription and translation proteins, and itself it shows interesting response in teh wild type to stress. This analysis nicely positions this gene as a central regulator in noise of yeast gene, as a mediator of bet-hedging. Strikingly this function was found to be conserved all teh way to s pombe.

This is one of the most elegant papers I have seen in years in teh study of noise of gene expression, and it provides one of the best results I have ever seen from experimental evolution! I congratulate the authors for super work. Everything they have done here was done masterfully, from the execution of the original evolution , to the very elegant segregant analysis, to their homing in on the regulators, and its characterization, and evolution. 

Most importantly, the paper addresses and solves an important challenge - find regulators of noise, and their mechanism of action! few papers before have done something of this scale.

I could not find any potential "complaint" or advice. It never happened to me before as a referee.

I could have advised the authors to look genome wide at the G70D mutant to see if it affects noise of other genes, or if the effect is pathways specific. But I think they have a rounded story already now, and they (or others now) may leave that for the next one. 

(I have blasted myself the protein againts otehr fungi and found that position 70 is very highly conserved, by that occasionally it's mutated (e.g. to Ala in Venturia inaequalis Colletotrichum fructicola), but never into an Asp as found here)

I thus enthusiastically recommend publication AS IS!

--

Reviewer #3: 

This study investigates the role a methyltransferase in the buffering of “noise” in yeast. By performing microbial evolution experiments on S. cerevisiae to select for mutations that increase reporter protein noise, combined with bulk segregant analysis and CRISPR/Cas9-based reconstitution, the authors identified a molecular regulator of noise, namely methyltransferase Hmt1. Noise regulation was found primarily to be achieved via two Hmt1 methylation targets. Furthermore, Hmt1 was also found to buffer gene expression noise in S. pombe, a distant relative of S. cerevisiae, suggesting generality of the noise buffering mechanism. Overall, this is a highly relevant and timely study that has important implications for the fields of cellular/molecular biology and epigenetics, as well as for other related fields such as antifungal resistance research. Overall, the experiments are rigorous and support the main finding, the manuscript is well written, and the figures are of good quality. I support publication in PLOS Biology once the minor concerns below are addressed. 

Minor Concerns

-There are several works in the literature that I would like to see cited: 1) A review article on gene expression noise written by Kaern et al., Nat. Rev. Genet., 2005, which would provide a comprehensive introduction for readers unfamiliar with expression noise and its consequences. 2) When specifically discussing gene network topology and noise (e.g., on Page 5, Line: 11), it would be appropriate to cite Charlebois et al., Phys. Rev. E, 2014; it would also be relevant to cite a related study demonstrating that the properties of gene expression noise can evolve on the fitness landscape (Charlebois, Phys. Rev. E., 2015), for example, on Page: 5, Line 5. 4) Very recently, Farquhar et al., Nat. Commun., 2019 demonstrated that gene expression noise modulated by gene network topology can influence drug resistance and the selection/establishment of genetic mutations (relevant, e.g., on Page: 4, Lines: 8-9) - it would be interesting to discuss this work in the context of your findings. 

-Page 4: Lines 13-16. Perhaps this is just semantics, but I think it would be more accurate to say “complex regulatory networks that respond to randomly fluctuating environments…”, because if the environment fluctuates periodically then, for instance, oscillating gene regulatory network dynamics (which seem to have selected for in other instances, such as circadian rhythms) would confer a lower fitness cost than both sense/response networks and bet-hedging. 

-Page 8: Lines 10-14. Be cautious about concluding that an observed phenomenon is nongenetic in nature from “relaxation” type experiments, as for example, it is possible that the mutants were instead outcompeted by “wild type” cells when the experimental conditions changed. The only way to ascertain for certain whether an adaptation is nongenetic or genetic is to perform sequencing (as you have done in the rest of the study). 

-Avoid overstating your findings. For instance, on Page 14: Lines 13-15 I would say “may represent an important survival strategy…” instead of “probably represents an important survival strategy…” in the context of “diverse microorganisms”. Alternatively, I think it would be reasonable to say “probably represents an important survival strategy in yeasts”. 

-Any scripts/codes used to analyze the experimental data and perform sequence simulation should be made is freely available on a site such as GitHub – this will ensure reproducibility of the results and maximize the utility of this work to the research community.

---

## [Editor Report · Decision Letter 2]

27 Sep 2019

Dear Dr Leu,

On behalf of my colleagues and the Academic Editor, Csaba Pál, I am pleased to inform you that we will be delighted to publish your Research Article in PLOS Biology. 

Early Version

PRESS 

Kind regards,

Sofia Vickers

Senior Publications Assistant

PLOS Biology

On behalf of, 

Hashi Wijayatilake,

Managing Editor

PLOS Biology